# On the Mechanical Performance of Polylactic Material Reinforced by Ceramic in Fused Filament Fabrication

**DOI:** 10.3390/polym14142924

**Published:** 2022-07-19

**Authors:** Lotfi Hedjazi, Sofiane Guessasma, Sofiane Belhabib, Nicolas Stephant

**Affiliations:** 1ESTP Campus de Troyes, 2 Rue Gustave Eiffel, F-10430 Rosières-prés-Troyes, France; lhedjazi@estp-paris.eu; 2INRAE, UR1268 Biopolymères Interactions Assemblages, F-44300 Nantes, France; 3Université de Nantes, Oniris, CNRS, GEPEA, UMR 6144, F-44000 Nantes, France; sofiane.belhabib@univ-nantes.fr; 4Nantes Université, CNRS, Institut des Matériaux Jean Rouxel, IMN, F-44300 Nantes, France; nicolas.stephant@univ-nantes.fr

**Keywords:** polylactic acid, ceramic particles, fused deposition modelling, tensile properties, 3D printing parameters

## Abstract

This study addresses the potential of using ceramics-based filaments as a feedstock material in an additive manufacturing process. Tensile specimens of PLA-ceramic (PLC) material are manufactured using a fused deposition modelling process, applying various printing parameters including printing angle and part orientation. Mechanical testing is performed on both the filaments and 3D-printed parts, and the related engineering quantities are derived. The experimental results show that PLC wire properties are substantially restored for the horizontal and lateral printing orientations, with only a 9% reduction in stiffness. In addition, a typical elastic-plastic response is achieved with these orientations, allowing the PLC to achieve excellent stiffness and elongation-at-break performance. The mechanical performance of the PLC is explained by the large proportion of continuous filaments along the loading direction. In addition, the printing angle is found to be a secondary factor allowing for layups at −45°/+45° and 0°/90°, resulting in the best tensile performance. The downside of using PLC is the lack of mechanical transfer, which is associated with weak interfacial behaviour and the inability to achieve high tensile strength.

## 1. Introduction

Additive Manufacturing (AM), also known as 3D printing, encompasses a wide variety of techniques that can be commonly defined as processes of joining materials layer by layer from a digitalised model [1,2]. These processes provide a unique framework to build materials with a large amount of complexity while decreasing the need for tool adaptation [3,4,5]. Under this scheme, AM is able to reach a high potential of technical part customisation [6,7]. In addition, it makes it possible to locally control material deposition depending on the needed functionality [8]. This has led, for instance, to the development of the functionally graded materials concept as a new major route for progress in AM [9,10]. Among the AM processes available, one has attracted attention for processing polymeric materials since the early development of this technology, namely, fused deposition modelling (FDM) [11,12]. This technique consists of laying down a melted filament via a moving printing nozzle. Among the feedstock filament materials that have been studied, polylactic acid (PLA) and acrylonitrile butadiene styrene (ABS) were on the top list [13,14]. For instance, early work by Ahn et al. [15] showed the importance of part orientation in achieving anisotropic behaviour in ABS printed materials. More recent work by Samykano et al. [16] considered a large number of processing parameters including infill rate, raster angle, and layer thickness. The authors concluded that significant parametric interdependency exists with regard to achieving high tensile performance of printed ABS. Lee et al. [17] studied the influence of the cooling intensity on the dimensional stability and mechanical performance of 3D-printed PLA. The authors demonstrated that the cooling efficiency triggered better dimensional stability but with the side effect of lowering the tensile strength. More recently, Czyżewski et al. [11] considered the influence of the nozzle diameter on the microstructure and performance of 3D-printed PLA. The authors showed that a nozzle diameter larger than 0.4 mm induced a large amount of process-induced porosity. In addition, the same authors highlighted the loss of mechanical performance when using a 0.2-mm nozzle diameter. In order to improve the performance of 3D-printed parts beyond prototyping applications, a research route using composite materials as a filler is considered [18]. On one hand, the development of composite feedstock materials for AM makes it possible to maintain a performance threshold while improving the environmental footprint. On the other hand, such materials have certain characteristics other than mechanical ones, enlarging the spectrum of applications of AM-based materials [19]. Filaments such as PLA-wood have been extensively studied. Cuan-Urquizo et al. [20] showed that the infill density as well as the topology (i.e., cell morphology) have a significant effect on the flexural properties of cell-like PLA-wood structures. Guessasma et al. [21] showed that PLA-wood is printable within a wide range of printing temperatures. However, a significant loss in tensile performance occurs, mainly in terms of stiffness and tensile strength, which limits the application of the PLA-wood material to prototypes. 

PLA-Ceramic has drawn attention as a means of replacing conventional metal-based materials used in surgery and bone replacement [22,23]. According to a review paper by Chen et al. [24], several AM routes are available for printing ceramic materials such as powder-based fusion, slurry-based photopolymerisation methods, and fused filament methods. These methods are not equivalent in terms of part rendering, generated residual stress, defect genesis, or surface finishing. The printing of ceramics is an ongoing research field according to a recent review paper by Lakhdara et al. [25]. Windsheimer et al. [26] considered SiC-filler-loaded pre-ceramic paper in a four-stage process to achieve sheet-laminated 3D printing of SiC ceramic. The authors reported process-generated porosities as small as 1 µm, attributed to interparticle separation. Ekel et al. [27] reported a stereolithography route for printing ceramic cellular materials using preceramic monomer ultraviolet light curing. The authors showed that this route leads to a smooth surface finish, zero-porosity, and higher performance compared to commercially available foams. Ahmed et al. [28] considered the fused filament route to study the mechanical performance of PLA/silica blends with contents ranging from 5% up to 15%. The authors demonstrated the an optimal content of ceramic, i.e., close to 10%, resulting in the highest tensile performance. In this study, the fused deposition route is used to print a PLA-ceramic blend. Although the literature contains several works describing successful printing using this technology [29,30], there is a lack of quantitative evaluations of the combined effect of the printing conditions, especially on the rendering of printed structures. Rendering is affected by several factors, such as the intrinsic properties of the filament and the anisotropy of mechanical behaviour. These, in turn, are affected by the amount of generated porosity and its spatial distribution. Printing angle and part orientation are among the printing conditions that have a strong effect on anisotropy. These are considered in this study and their combined effect on tensile performance is discussed.

## 2. Materials and Methods

Commercially available PLA-ceramic filament (PLC) was used as a basis for the experimental study. The filament was purchased from Frontierfila Company (Shenzhen, China) and had a ceramic particle volume content of 15%. The printing (T_P_) and base (T_B_) temperatures recommended by the supplier were as follows: T_P_ between 200 °C and 240 °C, and T_B_ varying between 50 °C and 80 °C. All 3D printing of PLC materials was performed on a Raise 3D Pro2 Plus printer. The printing conditions shown in Table 1 were combined to manufacture 3D-printed dogbone shapes (Figure 1) made of PLC filament. Three main sample orientations were selected, namely vertical, horizontal, and lateral (Figure 1a). 

These correspond to a building direction parallel to the length, thickness, and width of the sample. In addition, the printing angle corresponds to the arrangement of the filament within the plane of construction, which is normal for the building direction. In this study, four printing angles were selected: 0°, 15°, 30°, and 45°, generating layups of −45°/+45°, −30°/+60°, −15°/+75°, and 0°/+90°, respectively [31]. These layups were generated in the Cura software and are illustrated in Figure 1b–d for all three orientations, where the filament arrangement is fully captured by the two successive layers. The number of discontinued paths varied from one path to another. For instance, the horizontal orientation encompassed the largest filament paths (Figure 1d), whereas the vertical orientation only allowed short paths (Figure 1b).

The number of combinations reflecting parameter interdependency was found to be 12. With a minimum of two replicates per condition, the total number of printed samples was 24. Figure 2 illustrates the process for the three types of part orientations. It has to be mentioned that the lateral orientation required the addition of support material to avoid overhangs. The support was carefully removed by hand to avoid creating surface defects that could result in low tensile strength.

The other printing parameters were fixed at the ground values (Table 2). The support material was only used for printing the lateral configuration. 

Tensile experiments were performed on a 10 kN tensile/compression apparatus from Zwick Roell Group (Ulm, Germany) with a crosshead speed of 5 mm/min up to dogbone specimen failure (Figure 2). Tensile properties such as Young’s modulus $E_{Y}$, yield stress $\sigma_{Y}$, tensile strength $\sigma_{S}$, ultimate stress $\sigma_{R}$, and elongation at break $\varepsilon_{R}$ were measured from the stress–strain curve using an automated extracting algorithm. In order to monitor the sample deformation, a high-speed camera was coupled to the tensile setup (Figure 3). The Phantom V7.3 camera from Photonline company (Marly Le Roi, France) was used for this purpose. The measured area was adapted according to the testing conditions with a typical region-of-interest covering the full frame of the camera (800 × 600 pixels). However, other configurations were also used, down to 300 × 500 pixels. The physical size of each pixel is 150 µm. The camera acquisition speed was adjusted from 100 to 5000 frames per second (fps). The same experimental setup was used for testing the filaments in order to detect possible mechanical loss. 

Scanning Electron Microscopy (SEM) characterisation was conducted on both as-received filament and 3D-printed samples to study the chemical composition of the PLC filament as well as the fracture patterns as a function of the printing conditions. SEM observations were made using a JEOL JSM-5800LV (IMN, Nantes, France) operating with an accelerating voltage of 10KV. An Everhart-Thornley secondary electron detector was mainly used, as well as a backscattered electron detector for ceramic phase analysis. Energy dispersive spectrometry analyses were conducted with a SAMx SDD detector counting for 60 s under a beam current of 600 pA. The observable surfaces of the samples were coated using a Balzers CED 30 carbon evaporator that applied 50 nm of carbon to ensure conductivity during observations. Magnifications were adjusted from 25× to 16,000× with a typical pixel size ranging from 9 nm to 4 µm. 

## 3. Results and Discussion

### 3.1. Properties of As-Received PLC Filament

Figure 4a shows an SEM micrograph encompassing both longitudinal and cross-sectional views of an extruded PLC filament with dimensions of around 100 µm in width and 500 µm in length. The filament surface state was smooth compared to those of composite filaments such as wood-PLA or flax-PLA. The breakage along the cross-section showed a tearing effect (Figure 4b) but the contrast from the secondary electron imaging did not reveal the morphology of the ceramic particles.

A comparison between secondary electron imaging and Backscattered-Electron (BSE) imaging performed on the same large area (500 × 240 µm^2^) from the filament showed the presence of small globular particles (Figure 5). These particles, indicated with arrows in Figure 5b, appeared in light grey due to the presence of large atomic number chemical species, and their surface fraction (<2%) was small compared to the volume fraction of the ceramic (provided by the supplier). The magnification was insufficient to capture the morphology of these particles. Figure 6 shows two magnified views of PLC filament cross-sections under the two acquisition modes. Figure 6 shows that the second phase particles were wrapped inside the PLA matrix when observed under secondary electron imaging.

The backscattered-electron imaging mode revealed the morphology of these particles, which exhibited irregular shapes with a typical size of 1.6 ± 0.14 µm and a shape factor of 0.6 ± 0.2. Energy dispersive X-ray analysis with the beam focused on these particles revealed the presence of molybdenum, fluorine, silicon, chlorine, copper or zinc, depending on the particles. This suggests that the ceramic reinforcement may be constituted of cermet (MoSi) or oxides (SiO_2_). 

In order to have an idea about the chemical composition of PLC, Energy Dispersive X-Ray Analysis (EDX) was used on typical areas of 5 × 5 µm^2^ (Figure 7). For comparison purposes, the same analysis was performed on PLA filaments (Figure 7a). Differences in carbon and oxygen proportions were observed between PLA (Figure 7a) and PLC (Figure 7b), with the latter comprising significantly more carbon if we observe the relative heights of the peaks (carbon vs. oxygen). Due to the lack of precision of EDX in quantifying light elements such as carbon, especially for a surface topography such as that of the samples studied, the exact ratio between carbon and oxygen could not be accurately measured. However, the extra carbon in the PLC cannot be exclusively explained by measurement uncertainty. This difference in the carbon–oxygen proportion suggests differences in compositions and the involvement of carbon in the composition of the reinforcing ceramic phase, including SiC, MoC or MoC_2_. Figure 7c,d shows the obtained chemical analysis focusing on two particles (Figure 6). This analysis demonstrates the presence of heavier elements such as molybdenum, zinc, and copper as well as traces of fluorine, silicon, and chlorine. This suggests that the ceramic reinforcement may be constituted of cermet (MoSi) or oxides (SiO_2_), but more likely MoC_2_ if we consider the extra carbon found in PLC.

### 3.2. Tensile Behaviour of PLC Filament

Figure 8 shows the tensile response of the as received PLC. For the purpose of comparison, the typical tensile behaviour of PLA is also shown. Repeatability in the tensile behaviour of PLC wire was not achieved, as attested by the response of the four tested replicates

Part of the explanation for this comes from the observed sudden drops in tensile force, which may have been due to the interfacial effect. These events of local damage are also observed in other types of PLA composites considered as feedstock materials in FDM such as PLA-wood, PLA-hemp, and PLA-flax fibres [21,32,33]. Regarding the comparison between the performance of PLC versus PLA, Figure 8 shows that the reinforcement expected from ceramic particles was not observed because of the lack of mechanical transfer due to the sudden changes in PLC reaction force during tensile loading. For the highest ranked PLC, the gain in stiffness was only 10%. However, the grades of PLA in both wires may have differed significantly and did not allow us to draw a clear conclusion besides the observed sudden changes in reaction forces. On average, the variations in the stiffness, elongation at break, and tensile strength of PLC wire with respect to PLA wire were −27%, 42% and −7%, respectively. The observed gain in elongation at break can be explained by the accumulated damage which resulted from the sudden changes in reaction forces. This further extends the ability of the material to stretch due to the creation of material discontinuities.

### 3.3. Effect of Part Orientation

Figure 9 shows the cracking behaviour of PLC printed in the longitudinal direction (vertical orientation, HA) with a fixed printing temperature of 200 °C and a nozzle diameter of 0.4 mm. Within the plane of construction corresponding to the part cross-section, the effect of filament layup was considered by varying the printing angle from 0° to 45°. In this configuration, small filament paths were the main features influencing the tensile behaviour, and were associated with higher thermal cycling. Figure 9 suggests that despite the varied layups with the plane of construction, all configurations led to unstable cracking, i.e., transverse cracks were observed in nearly pure opening mode. The overall behaviour depicted as tensile stress/strain response is shown in Figure 10 for typical replicates. Figure 10 confirms the brittle behaviour of all tested conditions irrespective of the printing angle. A clear ranking of tensile response with respect to the printing angle could not be made. The highest response corresponded to a printing angle of θ = 45°, which exhibited a sequence parallel to the width and thickness of the part. Table 3 shows the engineering constants extracted from all replicates with the corresponding standard deviation. Yield and ultimate stress, together with tensile strength, could not be differentiated. Under these configurations, the highest trends were confirmed by the low and upper printing angle bounds, namely, θ = 0° and θ = 45°. Additionally, the loss in mechanical performance was substantial for PLC samples printed in the vertical orientation, i.e., −31%, −80%, and −97%, for Young’s modulus, tensile strength, and elongation at break, respectively. Such a mechanical loss can be explained by the lack of cohesiveness along the building direction. When the structure was subject to tensile loading, the necking effect that characterises the building sequence altered the mechanical transfer between the layers, revealing most of the anisotropy of the behaviour of PLC. This is a common result for FDM feedstock materials, and PLC is not an exception [34].

Figure 11 depicts the SEM micrographs of PLC printed in the vertical orientation (HA) and using a printing angle of 45°. The filament length was normal to the loading direction. A large degree of variability in filament diameter was observed, i.e., from 148 µm up to 277 µm (Figure 11a); this was attributed to the stretching of the filaments along their diameter. The average filament diameter, however, was still close to the imposed layer height (203 ± 55 µm). Close to the top edge, the sudden change in nozzle trajectory affected the filament alignment and morphology, and in some spots, the diameter increased to 480 µm. Far from the fractured surface, the compactness of the filament along the building direction did not show inter-filament residual porosity, which partly explains the ranking of the tensile response of PLC printed at 45° (Figure 10). Near the fractured surface, evidence of inter-filament decohesion can be seen, which, in some cases, led to a significant reduction in the cross section (Figure 11b). Surprisingly, the good adhesion between the adjacent filaments showed that filament rupture was induced by transverse pulling instead of complete filament decohesion (Figure 11b). 

Figure 12 depicts the cracking behaviour of PLC printed along the sample thickness (horizontal orientation, LO) at the same printing temperature, i.e., 200 °C, and nozzle diameter, i.e., 0.4 mm. The contrast in terms of characteristics was highlighted by the larger extension of the sample prior to rupture and the overall ductile behaviour. In addition, the cracking behaviour was rather slow and damage growth and accumulation were visible, irrespective of printing angle. Within the plane of construction formed by the sample width and length, the effect of filament layup was found to be significant. For a printing angle of 0°, the crack deviation towards −45° was clearly distinguished, allowing significant shearing to take place. A residual tension was observed at the end of the tensile test, due to the uniaxial deformation of the external frame. This residual deformation contributed to avoiding the sudden rupture of the specimen. For a printing angle of 15°, the same cracking behaviour was observed with a crack deviation following a lower angle of 30°. This shift in crack deviation was in line with the initial filament orientation within the layup. Further increasing the printing angle to 30° resulted in a layup which was closely oriented towards the transverse direction. This made it easier for the transverse cracks to propagate throughout the external frame. In addition, multiple cracking is likely to occur, as shown in Figure 12b. For a printing angle of 45°, there was a great chance to witness full transverse cracking; compared to the vertical orientation (Figure 9a), this was not unstable. 

Figure 13 shows the obtained tensile response for all printing angles. All conditions led to higher stress levels, i.e., well above those for samples printed using vertical orientation. In addition, all responses exhibited elastic-plastic behaviour with a localisation trend similar to the wire response (Figure 8) but to a lower extent. With regard to the effect of the printing angle on the tensile performance, the same observation made for the vertical orientation holds for the horizontal orientation. Both *θ* = 0° and *θ* = 45° seemed to rank equally. Table 3 summarises the engineering constants for all LO conditions. According to this table, a positive trend may be seen between the printing angle and Young’s modulus. This trend can be captured using a linear function such as:(1)$$E_{Y}\left(MPa\right)=590+4.23\times \theta \left({}^{\circ}\right);\;\;\;\;R^{2}=0.98$$

This is a surprising result, given that the best layup for improving the tensile properties of the print was found to be the one favouring filament crossing in −45°/+45° sequence. 

As a secondary effect, the increase of the printing angle slightly improved the tensile strength, yield, and ultimate stress, according to the following linear approximations:(2)$$\sigma_{Y}\left(MPa\right)=30+0.04\times \theta \left({}^{\circ}\right);\;\;\;\;R^{2}=0.89$$
(3)$$\sigma_{M}\left(MPa\right)=31+0.07\times \theta \left({}^{\circ}\right);\;\;\;\;R^{2}=0.72$$
(4)$$\sigma_{R}\left(MPa\right)=6+0.01\times \theta \left({}^{\circ}\right);\;\;\;\;R^{2}=0.72$$

The direct effect of the improvement in these engineering constants was the slight decrease of the elongation at break according to the same linear approximation:(5)$$\varepsilon_{R}\left(\%\right)=14-0.09\times \theta \left({}^{\circ}\right);\;\;\;\;R^{2}=0.31$$

In this last case, the linear trend was not obvious because of the large standard deviation, especially for *θ* = 0°. The samples printed using the horizontal orientation achieved superior performance compared to those using the vertical orientation. This was not a surprise, considering that the lack of cohesiveness along the thickness did not act against the load transfer in the loading direction. Although the crossing of the filaments within the plane of construction induced varied shear and tension deformation depending on the printing angle, the tensile response still achieved the highest ranking. This was confirmed by the limited loss of stiffness and yield stress with respect to the PLC wire. Indeed, the average variations of the tensile properties with respect to the PLC wire properties were as follows: −9%, −16%, −33%, −84%, and −83% for Young’s modulus, yield stress, tensile strength, ultimate stress, and elongation at break, respectively. This means that adopting the horizontal orientation restores 15% of the stretching capabilities and reduces the loss in both stiffness and strength by 21% and 47%, respectively, compared to the vertical orientation. 

Figure 14 shows typical SEM micrographs of PLC printed in the horizontal orientation (LO) and using a printing angle of 15°. The plane of construction according to the part orientation (Figure 1) encompasses the length and width of the specimen. Figure 14a shows the result of the filament arrangement with the filament sequence −15°/+60° altered near the fractured zone. Several deformation mechanisms can be observed; the first is the filament tearing, which is more related to the filament stretching at the external frame. Significant shearing within the raster was also observed, with a significant amount of interfacial cracking (Figure 14a). 

Outside the fractured area, the filament packing seemed to be unaltered, with nearly no gap within the raster and a lower variability in filament size. The necking, generally reported as a source of large gap generation between filaments, was limited. The transverse dimension normal to the layer height was about 400 µm, which means that the transverse aspect ratio of the filament was close to 0.5 (Figure 14b). A slight change in the filament morphology was observed, which is indicated by the arrowed waviness in Figure 14b.

Figure 15 shows the cracking behaviour of PLC printed in the lateral orientation (Figure 1) at a fixed printing temperature of 200 °C and using a nozzle diameter of 0.4 mm. Within the plane of construction comprising the length and thickness of the sample, the effect of filament layup was limited, and only significant stretching of the filaments in the loading direction was commonly observed for all printing angles. This stretching led to damage onset and slow growth within a limited section of the sample. In comparison to the vertical orientation, printing using either horizontal or lateral orientations did not lead to the complete failure of the samples.

Figure 16 quantifies the observed tensile response with the help of the stress–strain sketches. In addition to the elastic-plastic behaviour observed for all printing angles, similar to the case of horizontal orientation, the printed samples using lateral orientation achieved roughly the same elongation as the PLC wire. This result may be explained by the large amount of filament oriented in the loading direction combined with the fact that these filaments were longer compared to those in the earlier orientations. The same ranking seems also to prevail, with a printing angle of 45° providing the highest tensile response and reduced elongation at break.

The two intermediate printing angles (*θ* = 15°, *θ* = 30°) resulted in the largest stretching. As such, there were complete ruptures for all printing angles due to the residual force that persisted at the end of the tensile loading. 

Table 3 shows the engineering constants for all printing angles. A strong nonlinearity was observed between all tensile properties and the printing angle, which prevented the use of a simple linear fitting procedure; a second order polynomial fit was used instead. The following forms were thus obtained:(6)$$E_{Y}\left(MPa\right)=699-4.9\times \theta \left({}^{\circ}\right)+0.14\times \theta 2\left({}^{\circ}\right);\;\;\;\;R^{2}=0.96$$
(7)$$\sigma_{Y}\left(MPa\right)=35-0.23\times \theta \left({}^{\circ}\right)+0.007\times \theta 2\left({}^{\circ}\right);\;\;\;\;R^{2}=0.93$$
(8)$$\sigma_{M}\left(MPa\right)=37-0.3\times \theta \left({}^{\circ}\right)+0.008\times \theta 2\left({}^{\circ}\right);\;\;\;\;R^{2}=1.00$$
(9)$$\sigma_{R}\left(MPa\right)=7-0.06\times \theta \left({}^{\circ}\right)+0.002\times \theta 2\left({}^{\circ}\right);\;\;\;R^{2}=1.00$$
(10)$$\varepsilon_{R}\left(\%\right)=24+0.67\times \theta \left({}^{\circ}\right)-0.015\times \theta 2\left({}^{\circ}\right);\;\;\;\;R^{2}=0.99$$

The loss of performance of the samples printed using the lateral orientation was similar to that observed in the horizontal orientation. On average, this loss represented −9.8%, −6.1%, −28%, −82%, −63% for Young’s modulus, yield stress, tensile strength, ultimate stress, and elongation at break, respectively. Thus, this orientation represented the best printing option among all tested orientations.

Figure 17 compares the SEM micrographs of PLC printed in the lateral orientation (LA) with two different printing angles. (0° and 45°). Significant tearing was observed in both cases due to the alignment of the filament length with the loading direction. Due to the strong filament lateral contraction, inter-filament decohesion was promoted as a second deformation mechanism, leading to the development of interfacial cracking by opening mode along the loading direction. As the building direction was aligned with the sample width, stronger necking was noticed with printing angle 0° (Figure 17a), whereas for a printing angle of 45°, the filament compactness seemed to be unaltered, even close to the fractured area. The plane of construction according to the part orientation (Figure 1) encompassed the length and width of the specimen. Figure 14a shows the result of the filament arrangement with the filament sequence −15°/+60° altered near the fractured zone. Several deformation mechanisms can be observed. At the edges, rare events of sudden filament rupture were observed (Figure 17b), which may be attributed to the stress localisation close to the edges involving the frame structure. 

Figure 18 shows the cracking behaviour of PLC samples printed using the vertical orientation. These are the sole samples exhibiting unstable cracking, as pointed out in Figure 5. Figure 18 demonstrates that successive snapshots at a time difference of 200 µs allowed to partly determine the crack speed, which started at small levels at 30 m/s and extended beyond 50m/s. The cracking initiation sites were mostly located at the sample edges, even if the external frame provided a shield again transverse stress localisation. In Figure 18a, there is also evidence of double crack initiation; these cracks started growing at different speeds and merged inside the sample. The double crack initiation could be attributed to the multiple stress localisation that can be triggered from the misalignment of the sample with respect to the loading direction. The frame rate was insufficient to capture faster cracks, as shown in Figure 18b. According to this scheme, there was no obvious effect of the in-place printing angle on the cracking behaviour.

Figure 19 shows a compression between PLC and two feedstock materials, namely, PLA and PLA-PHA, printable at low temperatures (200 °C) [35,36]. The same printing conditions were used for all feedstock materials in the horizontal orientation and with a printing angle of 0°. In order to render the data more comprehensible, relative performance with respect to wire properties is considered.

PLC exhibited the highest stiffness among the tested feedstock materials. Indeed, while the average Young’s modulus of both 3D-printed PLA and PLA-PHA was close to 0.66 GPa, PLC was 1.35 higher than the other feedstock materials. However, as expected, PLC did not reach the same levels of tensile strength as PLA or PLA-PHA, i.e., 28% lower than PLA-PHA and 39% lower than PLA. In addition, PLC has the benefit of restoring most of the stretching of the wire, which makes it a good candidate for applications where stiffness and elongation at break are needed. Indeed, its elongation at break was found to be 25% higher than that of PLA and almost 135% higher than that of PLA-PHA.

Figure 20 shows the design of a five-axis robot arm and rendering of the printed parts using a combination of PLC and PLA feedstock materials. The prototype is made of 24 parts and contains three joints that enable various degrees of movement. The total amount of material needed is 935 g, with a hallow structure and 100% infill rate to improve the mechanical stability. The printing duration is 235 h for all parts combined. To withstand the flexural load, the filament arrangement along the arm axes calls for horizontal and lateral orientations, where both are nearly equivalent thanks to the circular symmetry. The amount of support was optimised at less than 2% of the total weight of the parts, also allowing easy manual removal. The robot arm prototype is capable of handling medium size objects of less than 1 kg. 

## 4. Conclusions

This study concludes that the tested PLA-ceramic (PLC) feedstock material in FDM does not achieve a reinforcing effect, even at a quantity of 15% ceramic particles. Three-dimensional printed PLC has a higher stiffness compared to PLA but exhibits a loss of stiffness regarding its wire properties. Part of the explanation for this is the weakening effect of the interface, which seems to be a challenge not only for the tested material but also for varieties of other composites such as PLA-wood, PLA-hemp, and PLA-flax. Part orientation has a drastic effect on the tensile behaviour, and PLC is not an exception. This is a common conclusion in FDM, where anisotropy generated by the layering effect limits the tensile performance in the building direction. While the vertical orientation restored only a fraction of the PLC wire properties, it also resulted in the lowest tensile ranking and brittle behaviour. On the other hand, lateral and horizontal orientations induced elastic plastic behaviour with the best restored wire properties. When combining the effect of part orientation with the in-plane printing angle, two conditions seemed to have the greatest influence on performance: printing angles of 0° and 45°. At these two extreme levels, the tensile performance was improved, irrespective of the part orientation. SEM evidence suggests that the varied compactness of the filament arrangement gave rise to the observed tensile behaviour, i.e., promoting several deformation mechanisms including filament tearing, interfacial decohesion, and interfacial cracking. Finally, the tested feedstock exhibited the best balance between stretching and stiffness compared to other materials such as PLA or PLA-PHA. This indicates that PLC may be considered as a feedstock material for printing technical parts such as robotic arms, as illustrated in the present study.

## Figures and Tables

**Figure 1 polymers-14-02924-f001:**
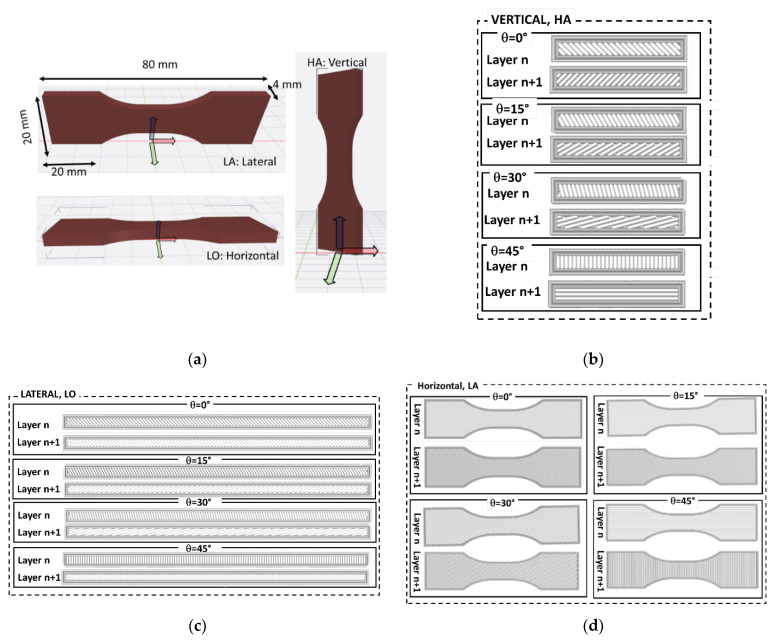
Illustration of the main printing parameters considered in this study: (**a**) Dogbone geometries and part orientation; and (**b**) printing angle combined with part orientation showing filament layups for vertical orientation; (**c**) layups corresponding to printing angles for lateral orientation; (**d**) horizontal orientation and related filament arrangement for all printing angles.

**Figure 2 polymers-14-02924-f002:**
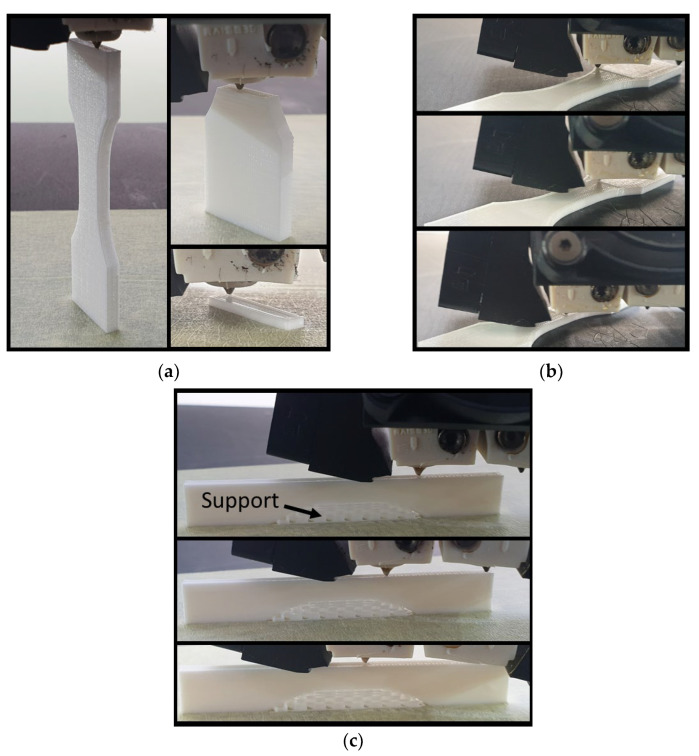
Ongoing 3D printing of PLC showing the three-part orientations: (**a**) vertical, (**b**) horizontal, and (**c**) lateral configurations.

**Figure 3 polymers-14-02924-f003:**
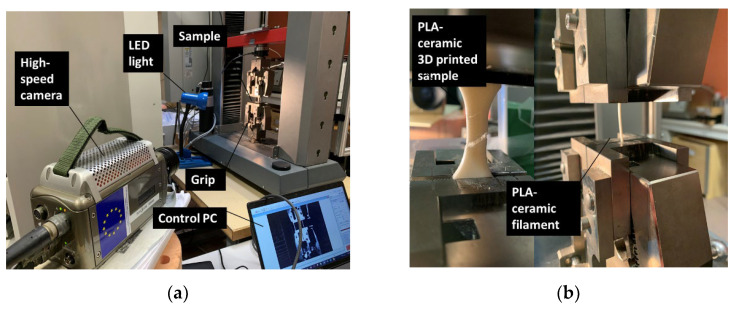
Tensile mechanical testing: (**a**) Experimental testing setup; (**b**) close-up view of filament and 3D-printed PLC sample.

**Figure 4 polymers-14-02924-f004:**
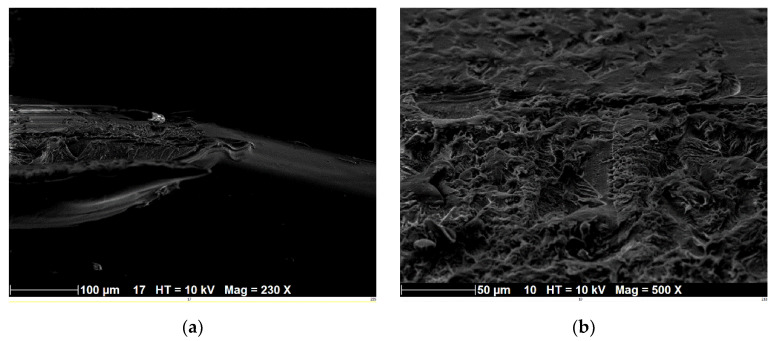
SEM imaging of PLC filament: (**a**) view along the filament length; and (**b**) close-up view of a filament cross-section.

**Figure 5 polymers-14-02924-f005:**
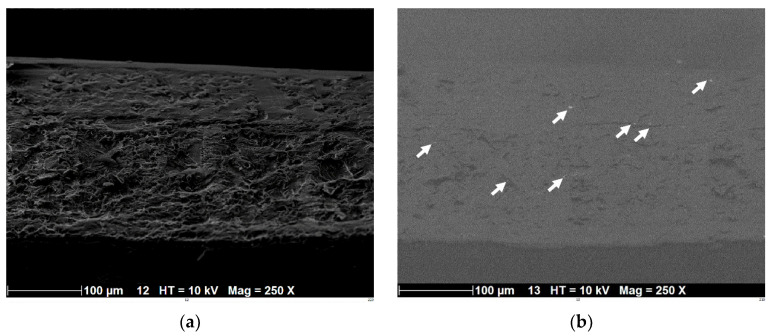
The same PLC cross-section view acquired using: (**a**) secondary electron imaging and (**b**) Backscattered-Electron imaging. The arrows point at the spots of ceramic particles composed of heavy elements.

**Figure 6 polymers-14-02924-f006:**
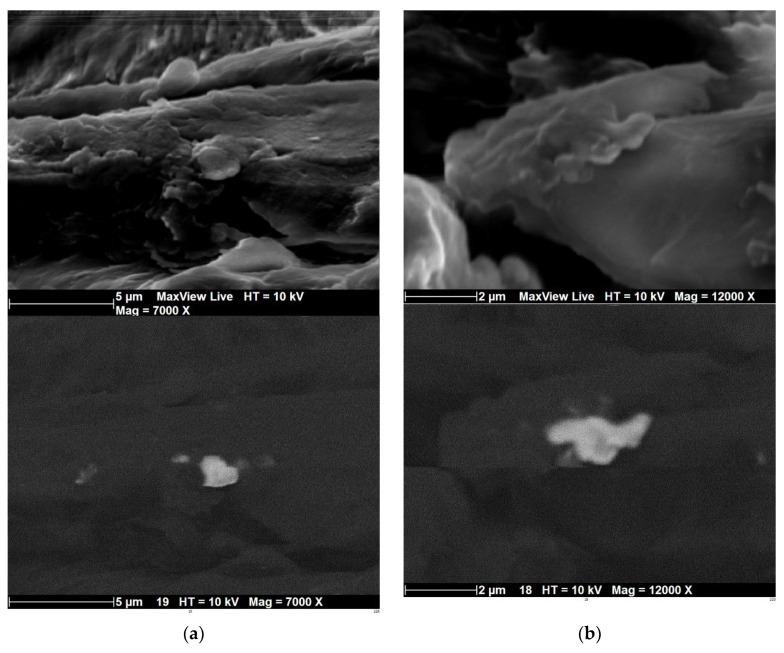
Zoomed-in views of a small area of a PLC filament cross-section showing the morphology of second phase particles using secondary electron imaging (top) and backscattered-electron imaging (bottom) modes with magnifications of (**a**) 7000× and (**b**) 12,000×.

**Figure 7 polymers-14-02924-f007:**
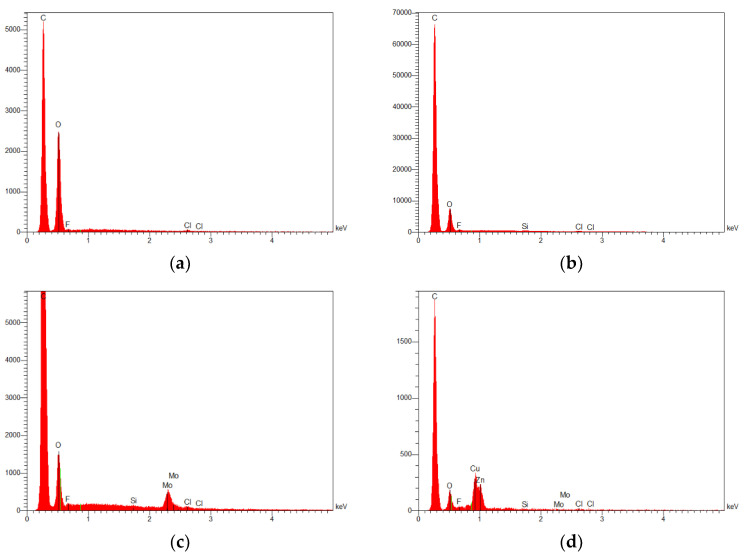
Energy Dispersive X-Ray Analysis (EDX) performed on large areas: (**a**) PLA and (**b**) PLC filaments, and focused on particles with typical areas of (**c**) 1.7 µm^2^ and (**d**) 2.2 µm^2^.

**Figure 8 polymers-14-02924-f008:**
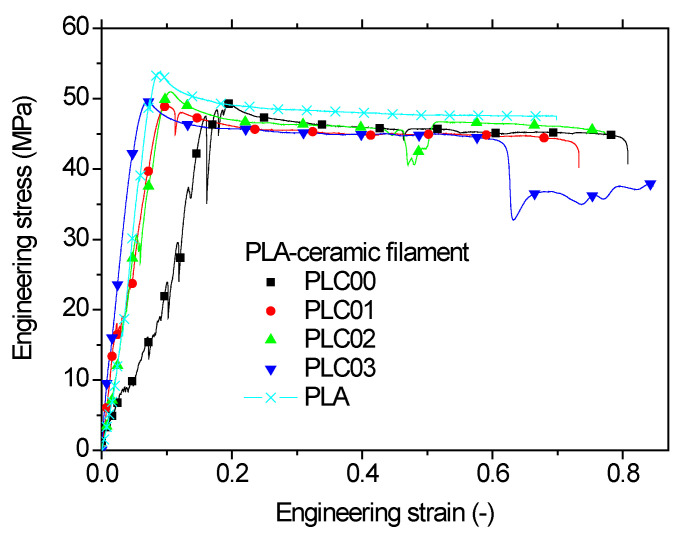
Comparison of the tensile response of PLC and PLA filaments.

**Figure 9 polymers-14-02924-f009:**
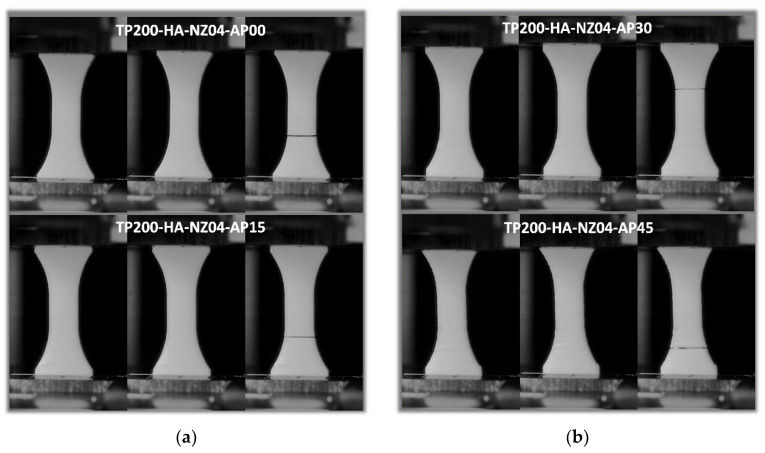
Effect of vertical orientation (HA) on the tensile behaviour of PLC for different in-plane printing angles: (**a**,**b**) close-up view of a filament and 3D-printed PLC sample.

**Figure 10 polymers-14-02924-f010:**
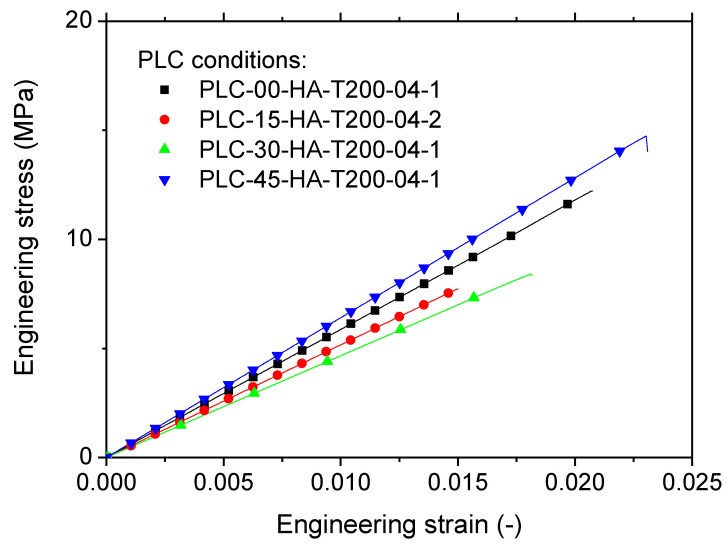
Tensile response of PLC printed using vertical orientation (HA) for various printing angles ranging from 0 up to 45°.

**Figure 11 polymers-14-02924-f011:**
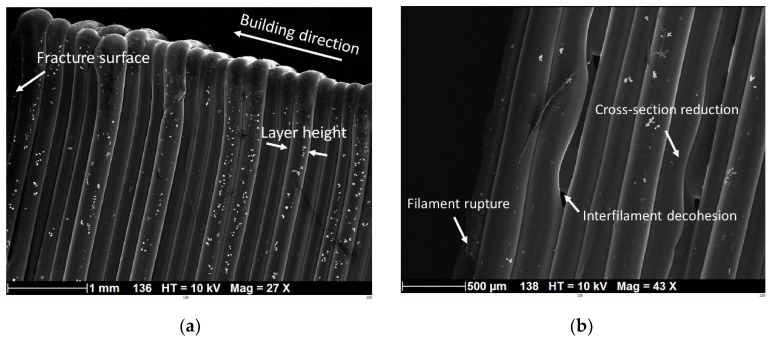
SEM micrographs of fractured surface of 3D-printed PLC according to vertical orientation (HA) (printing angle θ = 45°): (**a**) overall view normal to the sample thickness, and (**b**) zoomed-in view of the fracture surface.

**Figure 12 polymers-14-02924-f012:**
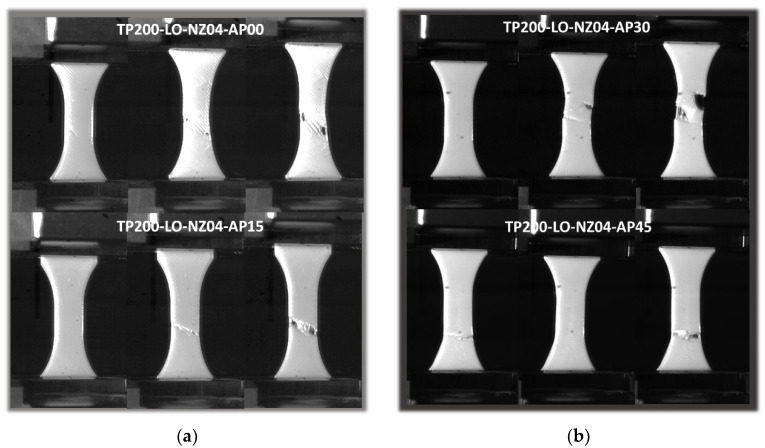
Effect of horizontal orientation (LO) on tensile behaviour of PLC for different in-plane printing angles: (**a**,**b**) close-up view of filament and 3D-printed PLC sample.

**Figure 13 polymers-14-02924-f013:**
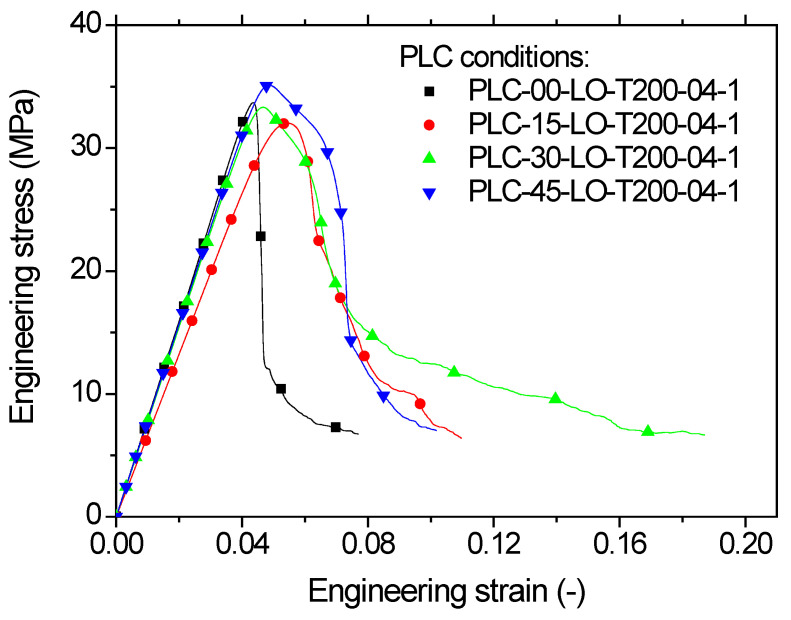
Tensile response of PLC printed using horizontal orientation (LO) for various printing angles, ranging from 0 to 45°.

**Figure 14 polymers-14-02924-f014:**
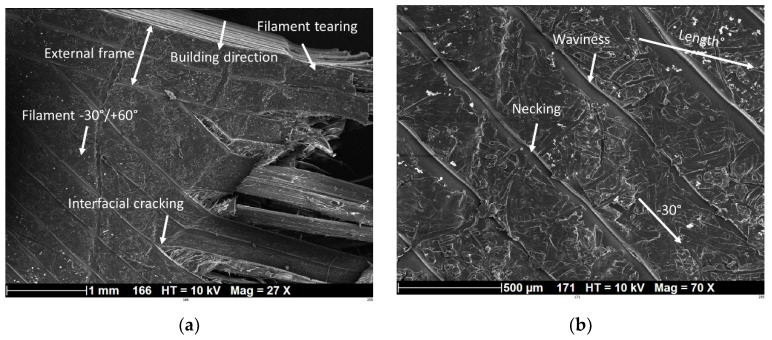
SEM micrographs of fractured surface of 3D-printed PLC according to horizontal orientation (LO) (printing angle θ = 15°): (**a**) close-up view on the fracture area, and (**b**) zoomed-in view of the raster.

**Figure 15 polymers-14-02924-f015:**
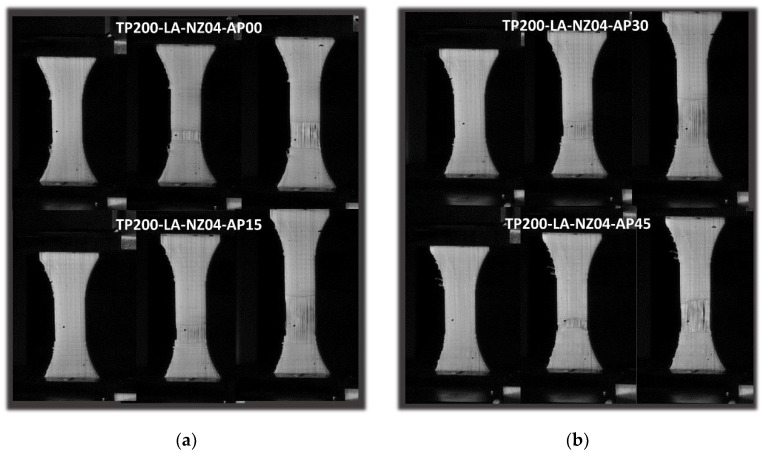
Effect of lateral orientation (LA) on tensile behaviour of PLC for different in-plane printing angles: (**a**,**b**) close-up view of filament and 3D-printed PLC sample.

**Figure 16 polymers-14-02924-f016:**
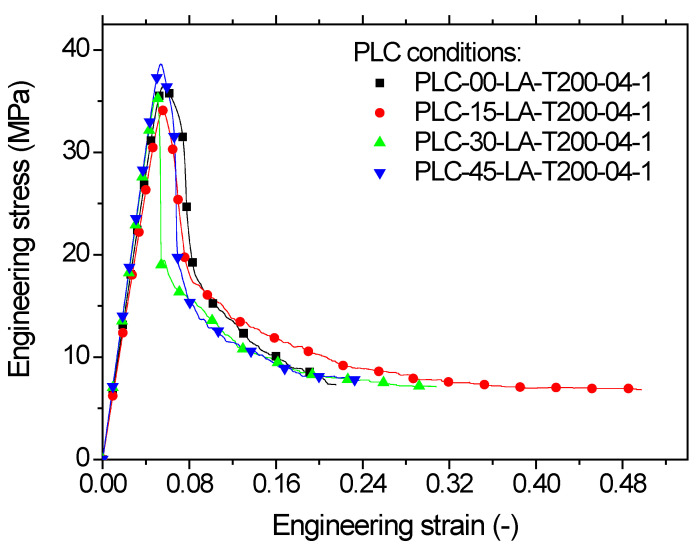
Tensile response of PLC printed using lateral orientation (LA) for various printing angles, i.e., ranging from 0 up to 45°.

**Figure 17 polymers-14-02924-f017:**
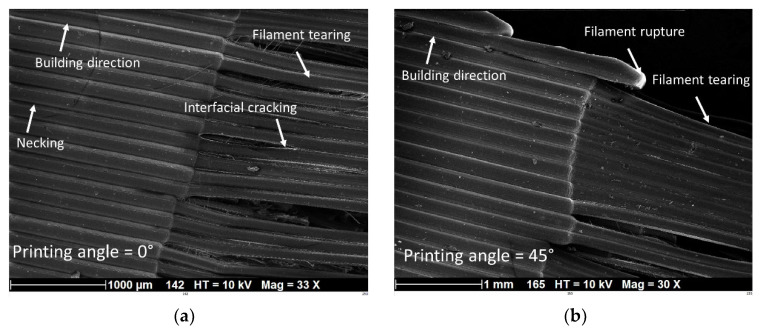
SEM micrographs of fractured surface of 3D-printed PLC according to lateral orientation (LA) with two printing angles: (**a**) 0° and (**b**) 45°.

**Figure 18 polymers-14-02924-f018:**
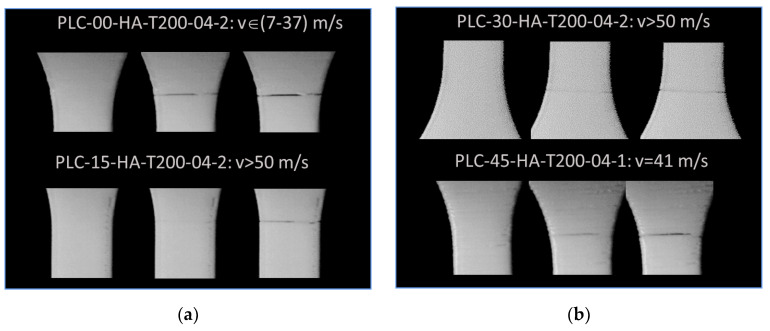
High-speed camera recording showing the crack behaviour of PLC printed using vertical orientation (HA) for different in-plane printing angles: (**a**) θ = 0°, θ =15°, (**b**) θ = 30°, θ = 45°.

**Figure 19 polymers-14-02924-f019:**
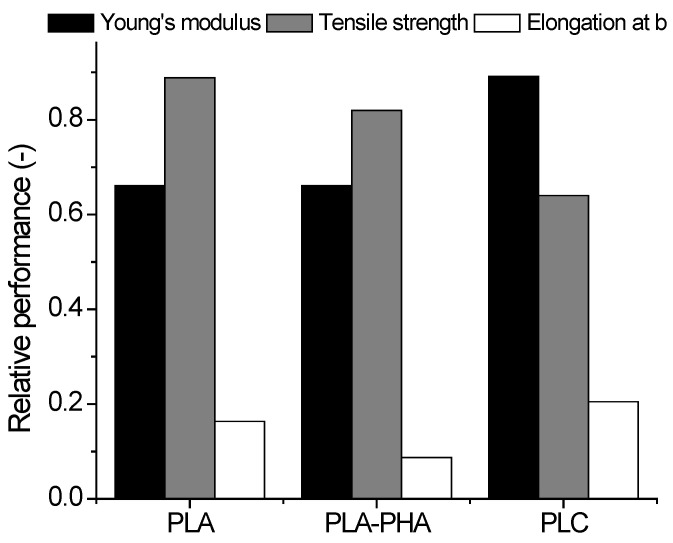
Comparison between the tensile performance of PLA, PLA-PHA, and PLC printed using horizontal orientation (HA) with a printing angle of 0°.

**Figure 20 polymers-14-02924-f020:**
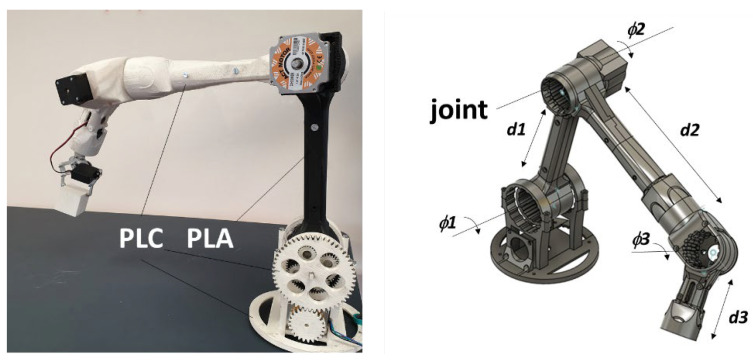
Design of technical parts used to build a robot mechanical arm using PLC with d1 = 29 cm, d2 = 30 cm, and d3 = 15 cm.

**Table 1 polymers-14-02924-t001:** Printing parameters used in this study.

Property	Level
Orientation, O_R_ (-)	Vertical (HA), Horizontal (LO), Lateral (LA)
Printing angle, A_P_ (°)	0, 15, 30, 45

**Table 2 polymers-14-02924-t002:** Fixed printing parameters used in this study.

Property	Level
Infill, I_F_ (%)	100
Nozzle diameter, D_N_ (mm)	0.4
Layer height (mm)	0.2
Printing temperature, T_P_(°C)	200
Bed temperature, T_B_ (°C)	60
Printing speed, V_P_ (mm/s)	50
Support density, S_PD_ (%)	10
Frame width, W_F_ (mm)	0.6, 0.8

**Table 3 polymers-14-02924-t003:** Observed tensile performance of PLC according to printing conditions.

Material *	E_Y_ (MPa)	σ_Y_ (MPa)	σ_M_ (MPa)	σ_R_ (MPa)	ε_R_ (%)
PLC wire	792 ± 167	37 ± 2	50 ± 1	41 ± 4	78 ± 5
PLC-00-HA-T200-04	596 ± 13	12 ± 1	12 ± 1	12 ± 1	2 ± 0
PLC-15-HA-T200-04	517 ± 11	8 ± 1	8 ± 1	8 ± 1	2 ± 0
PLC-30-HA-T200-04	487 ± 29	8 ± 1	8 ± 1	8 ± 1	2 ± 0
PLC-45-HA-T200-04	597 ± 62	13 ± 3	13 ± 3	12 ± 2	2 ± 0
PLC-00-LO-T200-04	706 ± 128	30 ± 5	32 ± 3	6 ± 1	16 ± 12
PLC-15-LO-T200-04	650 ± 16	31 ± 4	34 ± 2	7 ± 0	12 ± 1
PLC-30-LO-T200-04	735 ± 54	31 ± 1	33 ± 1	7 ± 0	17 ± 3
PLC-45-LO-T200-04	781 ± 5	32 ± 1	35 ± 1	7 ± 0	9 ± 1
PLC-00-LA-T200-04	699 ± 0	35 ± 1	37 ± 0	7 ± 0	24 ± 3
PLC-15-LA-T200-04	657 ± 3	32 ± 0	34 ± 0	7 ± 0	40 ± 14
PLC-30-LA-T200-04	735 ± 22	34 ± 0	35 ± 1	7 ± 0	30 ± 1
PLC-45-LA-T200-04	766 ± 9	38 ± 1	39 ± 1	8 ± 0	23 ± 1

* sample nomenclature: PLC-printing angle, orientation, printing temperature, and nozzle diameter; **E_Y_**: Young’s modulus; **σ****_Y_**: yield stress; **σ****_M_**: tensile strength; **σ****_R_**: ultimate stress; **ε_R_**: elongation at break.

## Data Availability

Data are available from the authors upon request.

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
