# Peer review of "On the Mechanical Performance of Polylactic Material Reinforced by Ceramic in Fused Filament Fabrication"

_polymers, 2022, doi:10.3390/polym14142924_

Round 1

Reviewer 1 Report

The manuscript entitled: “On the mechanical performance of polylactic material reinforced by ceramic in fused filament fabrication” is in line with the Polymer journal. The article based on original research. Overall, the article is well written, but it requires some changes before publication, such as the following:

·        Introduction – line 47: “Lee et al. [5]” wrong reference number.

·        Introduction: Information about the state of the art in 3D printing of material with ceramic reinforcement should be added.

·        Introduction (last paragraph): information on literature gaps should be clarified; the novelty aspects of the provided research should be provided.

·        Materials… (line 74): add more detailed information about the type of used ceramic.

·        Materials… (lines 119-128): please format the text properly (also other part of this article).

·        Figure 7: please comment the figure. Taking into consideration the methodology of samples preparation the investigation of carbon gives improper values.

·        Compare the results obtained with the literature more carefully.

Author Response

Reviewer 1:

Comments and Suggestions for Authors

The manuscript entitled: “On the mechanical performance of polylactic material reinforced by ceramic in fused filament fabrication” is in line with the Polymer journal. The article based on original research. Overall, the article is well written, but it requires some changes before publication, such as the following:

  • Introduction – line 47: “Lee et al. [5]” wrong reference number.

Sorry for the confusion. The right reference is now cited and renumbering of all references is performed.

Lee C-Y and Liu C-Y. Additive Manufacturing 2019;25:196-203.

  • Introduction: Information about the state of the art in 3D printing of material with ceramic reinforcement should be added.

We added a discussion on the printing of ceramic based materials starting by two review papers that provide a global picture of the available routes. Then we discussed some research achievement for some technologies emphasising on the key rendering of each technology in terms of surface finish, performance and defect presence.

Amendment in introduction section: “ According to the review paper by Chen…            the presence of an optimal content of ceramic close to 10% resulting in the highest tensile performance.”

The following references are added:

Chen Z, Li Z, Li J, Liu C, Lao C, Fu Y, Liu C, Li Y, Wang P, and He Y. Journal of the European Ceramic Society 2019;39(4):661-687.

Lakhdar Y, Tuck C, Binner J, Terry A, and Goodridge R. Progress in Materials Science 2021;116.

Windsheimer H, Travitzky N, Hofenauer A, and Greil P. Advanced Materials 2007;19(24):4515-4519.

Eckel ZC, Zhou C, Martin JH, Jacobsen AJ, Carter WB, and Schaedler TA. Science 2016;351(6268):58-62.

Ahmed W, Siraj S, and Al-Marzouqi AH. Polymers 2020;12(11).

  • Introduction (last paragraph): information on literature gaps should be clarified; the novelty aspects of the provided research should be provided.

The following paragraph is added to state the gap in literature and the novelty behind this study.

Added paragraph in introduction section: “Although, the literature shows several reported works leading to successful printing using this technology [29, 30], there is a lack of quantitative evaluation of the combined effect of the printing conditions, especially on the rendering of printed structures. This rendering is affected by several factors such as intrinsic properties of the filament and the anisotropy of mechanical behaviour. These, in turn, are affected by the amount of generated porosity and its spatial distribution. Among the printing conditions that have a strong effect on developed anisotropy is the printing angle and the part orientation.”

Added references

Onagoruwa S, Bose S, and Bandyopadhyay A. Fused Deposition of Ceramics (FDC) and composites. Proc Solid Freeform Fabricat Sympos 2001. pp. 224-231.

Rangarajan S, Qi G, Venkataraman N, Safari A, and Danforth SC. Journal of the American Ceramic Society 2004;83(7):1663-1669.

  • Materials… (line 74): add more detailed information about the type of used ceramic.

We are sorry to say that the only available information from the filament supplier is the amount of ceramic. This is reported in beginning of section 2. This is why we performed the SEM analysis to be able sense the ceramic particle morphology and composition. Detailed discussion of our results are given in section 3.1.

  • Materials… (lines 119-128): please format the text properly (also other part of this article).

Sorry for the formatting issue, this is now corrected and the entire manuscript is checked.

  • Figure 7: please comment the figure. Taking into consideration the methodology of samples preparation the investigation of carbon gives improper values.

The comment of Figure 7 is provided in the last version and the relative comparison between PLA and PLC is still valid. This is discussed in page 7 from the former version : “For comparison purposes, the same analysis is performed on PLA filaments (Fig. 7a). Differences in carbon and oxygen proportions are observed between PLA (Fig. 7a) and PLC (Fig. 7b). The latter comprising significantly more Carbon if we observe the relative heights of the peaks (Carbon vs Oxygen). Due to lack of precision of EDX to quantify light elements such as carbon, especially for a surface topography such as that of the samples studied, the exact ratio between carbon and oxygen cannot be accurately measured. However, the extra carbon in the PLC cannot be exclusively explained by the measurement uncertainty.”

In addition to this discussion, our comment on the difference in carbon intensity value is attributed to the presence of carbon in the composition of the ceramic phase.

We added the follwinng comment to support this statement:

“This difference in carbon-oxygen proportion suggests differences in compositions and involve-ment of carbon in the composition of the reinforcing ceramic phase including SiC, MoC or MoC2…. but more likely MoC2 if we consider the extra carbon found in PLC ”

  • Compare the results obtained with the literature more carefully.

We do not understand which results the reviewer is referring to. The comparison is provided only at the end of the results and discussion section through Figure 10. To make sure our comparison is meaningful, we used only two other types of filaments that were printed under the same conditions.

In order to provide more quantitative comparison, we added the following discussion for each of the three mechanical properties.

“Indeed, while the average Young’s modulus of both 3D printed PLA and PLA-PHA is close to 0.66 GPa, PLC is 1.35 higher than the other feedstock materials …. PLC reaches 28% lower than PLA-PHA and 39% lower than PLA… Indeed, its elongation at break is 25% higher than that of PLA and almost 135% higher than PLA-PHA. ”

Reviewer 2 Report

I have carefully analyzed the paper and I am really impressed by the cleariness, soundness and quality of the paper in terms of its scientific contect, graphical presentation, novelty, the way how methods, results and interpretation of these results are provided in this paper. I really appreciate that beside the characterizing / testing of the material presented in the paper by using specific methods (tensile testing, SEM, EDX analyses, etc) the authors are coming also with an application of printing of robot mechanical arm using PLA-ceramic (PLC) material.  References are up-to-date. I have realized also a checking with IThenticate application for similarity index analysis and the result achieved was really impressive in a good way. I find this paper as one that can be used as one good practice example / a reference paper and taking into consideration all these, I definitely recommend the acceptance of the paper entitled "On the mechanical performance of polylactic material reinforced by ceramic in fused filament fabrication" in this form to be published in the Polymers journal (MDPI). 

Author Response

Reviewer 2:

have carefully analyzed the paper and I am really impressed by the cleariness, soundness and quality of the paper in terms of its scientific contect, graphical presentation, novelty, the way how methods, results and interpretation of these results are provided in this paper. I really appreciate that beside the characterizing / testing of the material presented in the paper by using specific methods (tensile testing, SEM, EDX analyses, etc) the authors are coming also with an application of printing of robot mechanical arm using PLA-ceramic (PLC) material.  References are up-to-date. I have realized also a checking with IThenticate application for similarity index analysis and the result achieved was really impressive in a good way. I find this paper as one that can be used as one good practice example / a reference paper and taking into consideration all these, I definitely recommend the acceptance of the paper entitled "On the mechanical performance of polylactic material reinforced by ceramic in fused filament fabrication" in this form to be published in the Polymers journal (MDPI). 

We do really appreciate the comment of the reviewer and we thank him for looking in depth to our work.